# Monitoring and Surveillance of Small Ruminant Health in The Netherlands

**DOI:** 10.3390/pathogens11060635

**Published:** 2022-05-31

**Authors:** Eveline Dijkstra, Piet Vellema, Karianne Peterson, Carlijn ter Bogt-Kappert, Reinie Dijkman, Liesbeth Harkema, Erik van Engelen, Marian Aalberts, Inge Santman-Berends, René van den Brom

**Affiliations:** 1Department of Small Ruminant Health, Royal Animal Health Services (GD), P.O. Box 9, 7400 AA Deventer, The Netherlands; p.vellema@gdanimalhealth.com (P.V.); k.peterson@gdanimalhealth.com (K.P.); c.t.bogt@gdanimalhealth.com (C.t.B.-K.); r.vd.brom@gdanimalhealth.com (R.v.d.B.); 2Department of Pathology, Royal Animal Health Services (GD), P.O. Box 9, 7400 AA Deventer, The Netherlands; r.dijkman2@gdanimalhealth.com (R.D.); l.harkema@gdanimalhealth.com (L.H.); 3Department of Research and Development, Royal Animal Health Services (GD), P.O. Box 9, 7400 AA Deventer, The Netherlands; e.v.engelen@gdanimalhealth.com (E.v.E.); m.aalberts@gdanimalhealth.com (M.A.); i.santman@gdanimalhealth.com (I.S.-B.)

**Keywords:** monitoring, surveillance, components, small ruminant health, sheep, goat

## Abstract

In contemporary society and modern livestock farming, a monitoring and surveillance system for animal health has become indispensable. In addition to obligations arising from European regulations regarding monitoring and surveillance of animal diseases, The Netherlands developed a voluntary system for the monitoring and surveillance of small ruminant health. This system aims for (1) early detection of outbreaks of designated animal diseases, (2) early detection of yet unknown disease conditions, and (3) insight into trends and developments. To meet these objectives, a system is in place based on four main surveillance components, namely a consultancy helpdesk, diagnostic services, multiple networks, and an annual data analysis. This paper describes the current system and its ongoing development and gives an impression of nearly twenty years of performance by providing a general overview of key findings and three elaborated examples of notable disease outbreaks. Results indicate that the current system has added value to the detection of various (re)emerging and new diseases. Nevertheless, animal health monitoring and surveillance require a flexible approach that is able to keep pace with changes and developments within the industry. Therefore, monitoring and surveillance systems should be continuously adapted and improved using new techniques and insights.

## 1. Introduction

Globalization and climate change require an increased awareness that animal diseases can suddenly emerge worldwide. Well-designed and implemented monitoring and surveillance systems in livestock farming are important for early detection of disease outbreaks, to limit the impact on animal health and welfare, but, where appropriate, also on public health and food safety [1]. Within the past decades, various livestock monitoring and surveillance systems have been designed in which, although often interchangeably used, monitoring and surveillance complement each other well [2,3,4]. In this paper, monitoring is defined as the ongoing efforts directed at assessing animal health by a systematic collection and aggregation of information. Surveillance is defined as the action based on a signal or a deviation from a norm, following the interpretation of collected monitoring information.

Member states of the European Union (EU) have to comply with prevailing regulations in which general elements of monitoring and surveillance are integrated as described in the Animal Health Regulation (AHR) (Regulation EU no. 2016/429) [5]. In addition, the EU has established additional and disease-specific requirements for many diseases (Commission Regulation 2018/1882) [6]. According to the AHR, member states must delegate animal disease surveillance to a competent authority, nevertheless, are free in designing a system for collecting, comparing, and analysing relevant information regarding the disease situation. Consequently, most systems used are tailor-made and confined to control programs of listed diseases of FAO (Food and Agricultural Organization), OIE (World Organisation for Animal Health), and other (product) organisations and governments.

The Netherlands Food and Consumer Product Safety Authority (NVWA) is an independent agency of the Ministry of Agriculture, Nature and Food Quality (LNV), responsible for the supervision of notifiable diseases. However, with the sole use of disease-specific systems, a broad overview of trends in livestock industries is lacking, including insight into the occurrence and prevalence of new and emerging non-notifiable diseases and disorders. Furthermore, stakeholders have an interest in animal health and welfare at a high level, securing export positions, and public health protection by securing animal health and product safety.

Severe disease problems seen in cattle in 1999, related to the use of contaminated infectious bovine rhinotracheitis vaccine with bovine viral diarrhoea virus [7], led to the realisation that monitoring notifiable diseases alone does not provide sufficient information about the health status of livestock farming. Subsequently, a political motion was submitted by Waalkens and Ter Veer, two members of the House of Representatives of The Netherlands, who called on the government on the 9th of November 2000 to set up a central system for routinely reporting animal health-related information. This motion was the impetus for initiating a voluntary national monitoring and surveillance system for four major livestock production systems [4], including small ruminant health in The Netherlands in 2003. The objectives of the current system are (1) detecting outbreaks of designated animal diseases that are not endemic in The Netherlands, (2) detecting as yet unknown disease conditions, and (3) keeping track of trends and developments in small ruminant health in The Netherlands.

This paper describes this voluntary system of monitoring and surveillance in small ruminants in The Netherlands. A general overview of key findings and some elaborated examples of previous findings over nearly twenty years are reported, and the merits and future improvements are discussed. Another important aim of this paper is to exchange the knowledge gained and to support those countries that use or want to set up a comparable system.

## 2. Results

In a continuously changing world, livestock farming is adapting to changing circumstances and requirements. Therefore, systems that aim to monitor livestock farming, and in this case specifically small ruminant health, should also continuously be adapted and improved according to the changing requirements and circumstances. An overview of developments and improvements in the monitoring and surveillance system of small ruminants in The Netherlands, since its start in 2003, is presented in Figure 1.

The numbers of the annual phone and digital helpdesk consultations, post-mortem examinations, and farm visits together form the notifications with which animal health information is collected that can be transformed into key monitor indicators. Figure 2 gives an overview of these indicators for the period 2003–2021.

Small ruminant farms in The Netherlands have a seasonal lambing pattern with the majority of births in the first four months of the year. This is also reflected by the monthly number of helpdesk consultations (Figure 3A) and submissions for post-mortem examination (Figure 3B).

### 2.1. Overview of Findings

Key findings from the voluntary national small ruminant monitoring and surveillance system in The Netherlands (2003–2021).

Table 1 provides an overview of relevant findings from the monitoring and surveillance system of small ruminants in The Netherlands in the period 2003–2021. More detailed findings can be found in the published annual reports [8]. In this paragraph, the findings that are presented in Table 1 are explained briefly. Internal parasites like *Fasciola hepatica*, *Haemonchus contortus*, *Teladorsagia circumcincta*, *Dictyocaulus filaria*, and anthelmintic resistance play an important role throughout the evaluated period 2003–2021 [8,9,10,11,12,13,14].

The same applies to abortion, not only because of reproductive losses and economic consequences, but especially because of concerns for the farmer, their family, farm visitors, and, when *C. burnetii* is involved, people living in the surroundings [8,19,20,21,22,23,24,25,26,27,28,29,30,31].

Copper poisoning is a frequently recurring problem in many different sheep breeds but mainly in Frisian Milk Sheep, and occasionally in dairy goats. An increased copper concentration in compound feeds or milk replacer is a frequently found source of high copper concentrations [8].

Osteogenesis imperfecta was only seen once in four lambs on one farm [8]. Severe clostridial metritis is sometimes seen in sheep but more often in dairy goats, and occasionally there are dozens of cases on a farm in a short period of time [8].

The first outbreak of caseous lymphadenitis (CLA) in The Netherlands dates back to 1984 and was associated with imports of dairy goats [49]. A successful eradication program was implemented [50], although new imports of CLA infected small ruminants caused setbacks [8,38]. The dairy goat industry in The Netherlands is aiming for a situation in which only milk will be processed from CLA accredited farms.

Hyperoxaluria is a recessively inherited juvenile form of primary hyperoxaluria in Zwartbles sheep that was first reported during the Veterinary Laboratories Agency Weybridge meeting of the European Veterinary Surveillance Network in 2011, published as severe oxalate nephropathy in the same year [51], and found in The Netherlands afterwards until a breed-specific recessively inherited pathogenic alanine-glyoxylate aminotransferase variant was demonstrated as its cause [8,46].

Cerebrocortical necrosis in kids and dysbacteriosis associated diarrhoea in dairy goats are two probably related disease entities as a consequence of the way these animals are fed in the dairy goat industry, with relatively large amounts of concentrates, with chronic changes to the rumen as a consequence [8]. Research into these conditions is ongoing.

Spinal cord compression due to an injection site reaction following incorrect vaccine administration in the upper neck is regularly reported in individual sheep [8,52] and has been confirmed by post-mortem examination in dozens of animals in 2014, 2016, and 2017.

Although salmonellosis is a relatively uncommon disease in small ruminants, since 2016 several outbreaks caused by *Salmonella typhimurium* have occurred in dairy goat kids between one and four weeks of age, resulting in high mortality rates. On some farms, people also became ill, and the same relatively unknown multilocus variable-number tandem repeat analysis (MLVA)-type of this bacterium was demonstrated in animals and humans. The origin of this salmonella is unknown [8]. Research into this condition is ongoing.

In September 2018, ovine herpes virus type 2, the causal agent of malignant catarrhal fever, was demonstrated in liver tissue from a goat that was submitted for post-mortem examination after dying of vague symptoms and skin lesions. The goat farm of origin also held sheep. No further cases occurred [8]. Ovine enzootic nasal adenocarcinoma and ovine pulmonary adenocarcinoma or jaagsiekte are neoplastic diseases caused by the betaretroviruses enzootic nasal tumor virus and jaagsiekte sheep retrovirus, respectively [53]. These diseases have been confirmed in recent years, the former in a flock where, in 2018 and 2019, at least five cases were confirmed, and the latter in 2021, in an imported Scottish Blackface ram [8]. Both diseases are mentioned in this overview as examples of non-endemic but also non-notifiable diseases in The Netherlands. Although these diseases should preferably remain non-endemic, under current regulations, only voluntary disease control measures are possible. Osteomyelitis is a seldomly reported disorder in kids that has been confirmed for the first time in The Netherlands in 2018 [8,47] but has since been identified several times. Research into the background of this disorder is ongoing.

Pithomycotoxicosis is a hepatogenous photosensitisation in grazing ruminants caused by the intake of sporidesmin-containing spores of the saprophytic fungus *Pithomyces chartarum* that was confirmed for the first time in sheep in The Netherlands in 2019 [8,48]. Further cases were confirmed in 2020 and 2021. Research into climatic conditions in relation to these outbreaks is ongoing.

Floppy kid syndrome is a not well understood metabolic acidosis in goat kids characterized by wobbly gait, decreased muscle tone, anorexia, apathy, and depression, mainly starting between 4 and 14 days of age [54]. In most cases, only a few kids per herd are affected, but severe outbreaks with dozens of cases per herd do occur, and in those cases, indications have been found that the origin of this problem arises directly after birth [8].

### 2.2. Elaborated Examples of Three Findings

Outbreaks of bluetongue, Schmallenberg virus disease, and *Coxiella burnetii*-related abortions in dairy goats in The Netherlands have been detected by the voluntary monitoring and surveillance system. The case description of these outbreaks provides a detailed overview of how the monitoring and surveillance system functions.

#### 2.2.1. Detection of the Incursion of a Known Disease: The Case of Bluetongue

In August 2006, two sheep farmers in the southern part of The Netherlands contacted the GD consultancy helpdesk because of increased morbidity and mortality in their sheep flock (Table 1). A farm visit took place on Monday, August 14th as the cause of these problems was unknown [32]. Clinical signs indicated a suspicion of bluetongue, and follow-up actions were implemented; blood samples were collected and tested for the presence of the bluetongue virus, and animal movement restrictions were put in place. On Tuesday the 15th, the Central Veterinary Institute, now called Wageningen Bioveterinary Research (WBVR), reported positive PCR and serological test results. Two days later, on the 17th of August 2006, the Pirbright Institute, the EU reference laboratory for bluetongue, confirmed the results, and The Netherlands was declared bluetongue-infected. Measures were taken to reduce further spread [55].

On 28 August, the bluetongue virus isolated in The Netherlands was identified as bluetongue virus serotype 8 (BTV-8). On 18 August and 21 August, Belgium, and Germany, respectively, reported BTV infected ruminants. Retrospective epidemiological analyses indicated that the location of the first BTV infection was likely in Belgium [56,57]. However, to date, the route of the introduction of BTV-8 into north-western Europe remains unclear.

In July 2007, BTV-8 re-emerged in The Netherlands [58] and infected thousands of cattle and goat herds and sheep flocks. The bluetongue-associated mortality in sheep was estimated to be 27,000 in the second half of 2007 [34,59]. After a successful voluntary vaccination program with an inactivated vaccine, seroprevalence levels of BTV-8 in the Dutch sheep and cattle population increased to over 80% by the end of 2008 [60]. In the subsequent three years, large serological surveys were performed in The Netherlands in which no indications were found that BTV was still circulating [61,62]. Based on these findings, the European Commission reassigned the BTV free status to The Netherlands, on 15 February 2012.

#### 2.2.2. Detection of an Unknown Disease: The Case of Schmallenberg Virus

In December 2011, congenitally malformed lambs were submitted for post-mortem examination after phone contact with the GD consultancy helpdesk. Malformations included arthrogryposis, torticollis, scoliosis, kyphosis, brachygnathia inferior, and mild-to-marked hypoplasia of the cerebrum, cerebellum, and spinal cord [63]. Brain samples from malformed lambs were PCR positive for Schmallenberg virus (SBV), a novel arthropod-borne orthobunyavirus that was first discovered in mid-November 2011 in dairy cattle presenting with short febrile episodes, a drop in milk yield, and diarrhoea [39]. After confirmation of this first epizootic outbreak of congenital malformations in sheep and goats in Europe [40], the Dutch government decided at the end of 2011 to make it a notifiable disease with the aim of collecting scientific information about this new virus. A large research portfolio started immediately, with the intention of answering the question of whether this virus was zoonotic and blood samples were taken from 301 presumably highly exposed persons. No SBV-neutralizing antibodies were found in the serum of these participants, and therefore it was concluded that the public health risk for SBV was either absent or extremely low [42]. In 2012, it became clear that the disease had spread rapidly over Europe [64]. Further studies revealed high seroprevalences in sheep flocks in The Netherlands. Higher numbers of stillborn lambs, pre-weaning lamb mortality, repeat breeders, and lambs with abnormal suckling behaviour were observed significantly more often in case flocks compared to control flocks. However, the association between SBV infection and indicators such as mortality and reproductive performance seemed to be limited [44]. Based on the fact that vector-borne transmission by various *Culicoides* spp. can take place, further outbreaks were expected in seronegative animals [45], and indeed malformed lambs are born regularly even now, (Table 1) mainly in offspring of early mated seronegative animals.

#### 2.2.3. Monitoring Trends and Developments: The Case of *Coxiella burnetii* Infections

In 2005, *Coxiella burnetii* was detected as the causal agent in submissions of aborted materials originating from two dairy goat farms [19]. Although Q fever was first diagnosed in The Netherlands in 1956 [65], *C. burnetii* had not previously been identified as a cause of abortion in small ruminants in The Netherlands. These findings were shared with stakeholders of the monitoring and surveillance system, due to the possible risk to public health. Between 2005 and 2010, *C. burnetii* was found to be the cause of abortion in 28 dairy goat, and two dairy sheep farms [23]. Shedding of *C. burnetii* by dairy goats resulted in the largest human outbreak of Q fever ever recorded, with more than four thousand human cases [66]. In 2008, *C. burnetii*-related abortion in small ruminants became notifiable, and a large package of measures, aiming at reducing shedding and environmental contamination, was implemented. These combined measures finally resulted in the end of this outbreak [23]. Bulk tank milk surveillance [21] and an effective compulsory vaccination programme with the phase 1 vaccine Coxevac^®^ [28] are still applied on commercial dairy sheep and dairy goat farms [67].

## 3. Discussion

The Netherlands has a strong veterinary infrastructure with well-trained veterinary practitioners and second- and third-line veterinary organizations such as GD, the Veterinary Faculty at Utrecht University and WBVR, the NVWA as an independent agency responsible for notifiable diseases, and a modern livestock industry that knows how to make use of the aforementioned veterinary knowledge to improve animal health and welfare. Unfortunately, even under these apparently optimal conditions, a delay in noticing a veterinary problem or a hitherto new disease occurred several times before the implementation of the monitoring and surveillance system. An example of the latter was the severe disease problem in cattle in 1999 related to the use of bovine viral diarrhoea virus-contaminated infectious bovine rhinotracheitis vaccine [7]. This was the impetus for a national monitoring and surveillance system for all major livestock production systems [4], and for small ruminants in 2003. This task is performed by GD, an organization founded in 1919 that initially aimed at combatting bovine tuberculosis and brucellosis [68]. Since its establishment, GD has developed into an organization that provides animal health expertise, conducts diagnostic tests, carries out research projects, and coordinates several (voluntary) animal disease control programs. For all the tasks that are conducted by GD, the company is operating at the interface between livestock stakeholders, veterinary medicine, public health, and the government [69].

Since its initial development in 2003, GD continuously improved the monitoring and surveillance system to the changing requirements and circumstances as presented in the timeline in Figure 1. Although the basic design and aims have largely remained unchanged, the current system covers all the surveillance strategies as described by Hoinsville [70]. The consultancy helpdesk and diagnostic pathology service aim for early warning surveillance, passive surveillance, and participatory surveillance. The diagnostic laboratory component provides the possibility for early warning, passive surveillance, and risk-based surveillance [4]. To be able to monitor trends and developments, keeping track of a number of key indicators associated with small ruminant health as well as potentially influential factors, an annual data analysis is conducted in which routine census data are used to present indicators like animal density, mortality, and numbers of purchased and imported animals [71,72]. A similar system is in place for cattle [4].

The number of helpdesk consultations, post-mortem examinations, and farm visits fluctuates over time (Figure 2). These fluctuations can only partly be explained. There is no explanation for the decrease in the number of helpdesk consultations in the period 2003–2005. The increase after 2005 is mainly due to the Q fever outbreak from 2005–2010, the bluetongue outbreak in the years 2006–2008, and Schmallenberg virus disease in 2011 and 2012. Submissions for post-mortem examinations are voluntary and the decision to submit a carcass is mainly based on perceived value by the farmer and associated veterinary practitioner. In a survey of American dairy cattle veterinarians [73], reasons to perform a post-mortem included multiple animals affected, unexplained death, and diagnosis confirmation. In addition to economic considerations [74], social and cultural factors such as familiarity with a disease or fear of government involvement [75], and special requests in case of increased disease attention or disease outbreaks like Schmallenberg virus disease in 2011 [39] can also be part of the decision to submit an animal for post-mortem examination or not. The decline in the number of post-mortem examinations between 2007 and 2010 is possibly related to *Coxiella burnetii* abortions in dairy goats and dairy sheep, Q fever in humans, and restrictions being put in place as a consequence. Although the cause has not been extensively investigated, sheep and goat farmers and their representatives have indicated in various meetings that fear of possible consequences has caused farmers to hesitate to submit animals for post-mortem examinations [8]. The fluctuating numbers of farm visits depend on several factors, and since 2018, only farm visits with a direct link to the monitoring and surveillance system are reported.

The number of helpdesk consultations and post-mortem examinations also fluctuates during the year (Figure 3). These fluctuations reflect events and infection patterns that occur during a year, for instance, the lambing season, and subsequently the growing season of lambs, and the infection cycle of parasites such as *Haemonchus contortus* and *Fasciola hepatica*. In years of higher prevalences, submission numbers increase.

Bluetongue is an example of a disease in which the consultancy helpdesk played a pivotal role in its early detection. Although the virus had already circulated in Belgium [55,57], its first detection in The Netherlands indicates the added value of having a monitoring and surveillance system in place. In the case of bluetongue, the early detection led to the rapid implementation of regulations that reduced the speed of transmission in 2006. This provided the opportunity to study the epidemics of a known disease not previously identified in north-western Europe, and to gain a lot of knowledge on clinical signs, prevention, and treatment. Unfortunately, the virus returned in 2007 and spread rapidly almost acting as a directly transmittable disease [76]. Again the monitoring system supported stakeholders, veterinarians, and farmers by continuously providing information. Additionally, analysis of routine census data indicated that, in 2007, 34,853 sheep farms were registered in the identification and registration (I&R) database in The Netherlands and 3246 of them had officially notified a bluetongue outbreak in that year. The estimated seroprevalence of BTV-8 exposed locations in 2007 was 70% for sheep (95% CI: 63–76%), with a median within-location seroprevalence on BTV-8 infected locations of 67%. The bluetongue-associated mortality in sheep was estimated to be 27,000 in the second half of 2007 [34,35,59,77] and enabled quantifying fertility issues and mortality associated with bluetongue infections in cattle on a national level [78]. As part of the active monitoring system, since 2013, an annual risk-based screening has been conducted in cattle, given that cattle are the preferred host of *Cullicoides* spp. In the period between 2013–2020, all submitted samples tested negative and freedom from infection was proven. Since the implementation of the Animal Health Law (EC Regulation 2016/429) [5] in April 2021, the status of bluetongue changed from a disease that required immediate eradication to a disease listed as a category C disease (EC regulation 2018/1882) [6] i.e., a disease that may be subject to optional eradication programs. This change in regulation did not alter the monitoring scheme in place.

In 2015, the presence of BTV-8 was reported in France. As a consequence, a risk-based assessment was made for the introduction of bluetongue in The Netherlands. The risk of BTV-8 re-introduction was classified as substantial. Nevertheless, no additional actions were taken. In the past years, BTV-8 has been found in several European countries resulting in continuous awareness of potential reintroduction. Therefore, stakeholders of the Dutch monitoring and surveillance system are informed of the actual bluetongue picture in Europe every quarter, and immediately if necessary.

Detection of Schmallenberg virus disease in 2011, until its introduction an unknown disease, is an example of the second objective of the monitoring and surveillance system. Once again, the consultancy helpdesk played a major role in recognising an unusual situation. Post-mortem examination of congenitally malformed lambs proved SBV positive, using a recently developed PCR after detection of the disease in dairy cattle only weeks earlier. For a short period, Schmallenberg virus disease was made notifiable by the authorities, to be able to collect information and assess its zoonotic potential [42]. GD continues to passively monitor SBV via the consultancy helpdesk and pathological examination, and is vigilant on the possible emergence of other related arthropod-borne and possibly zoonotic agents, like Cache Valley fever virus [38], causing congenital arthrogryposis-hydranencephaly syndrome.

The Q fever outbreak in The Netherlands is an example in which the third objective of the monitoring and surveillance system was met. It shows the importance of the different surveillance components and their mutual cooperation in the exchange of information. After contact with the consultancy helpdesk, abortion in goats caused by *Coxiella burnetii* was confirmed in kids, and placentas that were submitted for post-mortem examination in 2005 [19]. Subsequently, stakeholders were informed. The impact of this disease became obvious when the number of human Q fever patients increased from an average of 17 per year between 1978 and 2006 to 168 in 2007 [79], and ultimately to more than 4000 human cases [80]. Collaboration between human and veterinary advisory boards lead to a national approach, aiming to reduce shedding and thus environmental contamination and as a consequence human exposure. Strengthened by this outbreak, there was a renewed focus on the causes of abortion as many of the infectious causes of abortion in sheep and goats have zoonotic potential [17]. However, the number of abortion submissions from small ruminants for pathological examination is still relatively low, and results potentially may not be representative of the causes of abortion in the total small ruminant population in The Netherlands [25,81]. To encourage farmers to investigate the cause of abortion, an easily accessible tool, based on deep throat swab sampling was developed [82] and implemented within the current monitoring and surveillance system for small ruminants. Farmers selected to participate in the annual surveillance to maintain the national *Brucella melitensis*-free status [39], were offered the opportunity to submit a deep throat swab sample from aborted or stillborn lambs or kids from 2022.

The overview of Table 1 clearly shows that the consultancy helpdesk, operated by diplomates and residents of the European College of Small Ruminant Health, plays a major role in the monitoring and surveillance system of small ruminants in The Netherlands. Expertise and interest in small ruminant health are key factors to adequately respond to the wide variety of consultations received by the helpdesk. Post-mortem examinations, performed by certified veterinary pathologists, are almost always necessary to confirm suspicions or play an important role in unravelling the background of many of the listed diseases. The quality and reliability of the system also depend on the data-analysis component that relies on datasets that need to be sufficiently representative for the small ruminant population. In order to increase the amount of data, new ways of collaborating with farmers and veterinarians are continuously being explored. For example, recently a collaboration between GD and veterinary practitioners was developed aimed at data collection of ‘field necropsies’ meanwhile supporting the practitioners with input from a pathologist. Under field circumstances, occasionally it might be impossible to submit deceased animals for pathological examination. Veterinarians who choose to perform a post-mortem examination themselves can now submit pictures, written observations, and selected samples for further testing which are reviewed by a pathologist. In turn, the submitted information and the results can be used for monitoring.

Monitoring and surveillance remain the works of men. Creating clear frameworks can help, however, the intention of individual persons to make it work is essential, not only within but also outside an organization such as GD. That is why networks like the Knowledge Network of Veterinary Practices, the Signalling Forum Zoonoses, and the European Veterinary Surveillance Network are essential as an integral part of this system.

## 4. Materials and Methods

The monitoring and surveillance system for small ruminant health has been performed by GD since 2003 and is supported and funded by both public and private stakeholders. Since April 2021, this system has been conducted as a statutory task by GD. It comprises ongoing efforts directed at assessing animal health by systematically collecting and aggregating information, and actions based on a signal or a deviation from the norm, after interpretation of collected monitoring information. Results are reported to stakeholders and subsequently communicated to the industry and other interested parties (Figure 4).

### 4.1. Small Ruminant Population

The identification and registration of sheep and goats in The Netherlands was gradually introduced in 1994, to provide better insight into the number and location of small ruminants. Taking into account an underestimation of the number of animals due to incomplete registration, this introduction resulted in the registration of 1,155,361 sheep and 617,647 goats at 46,427 unique small ruminant herds (UHI) in 2002 [83]. Since January 2010, it is mandatory to provide all small ruminants with electronic ear tags and to register them in a central identification and registration database (I&R), which contributes to the accuracy of animal registration [84]. In 2012, the small ruminant population consisted of 1,334,252 sheep and 431,068 goats at 39,275 UHIs, whereas in 2020 1,219,963 sheep and 670,842 goats were registered at 39,336 UHIs [81,85].

### 4.2. Collection of Information

To enable animal health assessment, the small ruminant monitoring and surveillance system in The Netherlands comprises non-disease-specific passive and active surveillance components in which a broad spectrum of signals are collected. Passive surveillance components rely on voluntary input from farmers, veterinary practitioners, and other livestock professionals who contact GD for an expert opinion on animal health problems they daily encounter. For the active surveillance components, GD takes action and collects data or samples in order to be able to investigate a specific subject. The current monitoring and surveillance system comprises four main components: (1) the consultancy helpdesk, (2) diagnostic services are passive surveillance components, (3) veterinary and international networks contain both passive and active parts, and (4) the routine analysis on census data is an active surveillance component.

#### 4.2.1. Helpdesk

The consultancy helpdesk is operated by diplomates and residents of the European College of Small Ruminant Health Management. The main aim of this helpdesk is to provide free specialist advice to veterinarians, farmers, and other livestock professionals who can either be contacted by phone or email. If desired or necessary, an on-farm visit can be arranged. These above-mentioned elements are regarded as the basis of this monitoring and surveillance system. Information derived from helpdesk contacts is reported anonymously and is never publicly traceable to the individual farmer or veterinarian. In the case that observed clinical signs reported to the helpdesk indicate a notifiable disease, contacts are referred to the NVWA.

#### 4.2.2. Diagnostic Services

GD provides extensive veterinary diagnostic laboratory services for both infectious (immunological, parasitological, virological, and bacteriological) and non-infectious diseases (chemical).

In addition, GD provides a diagnostic pathology service operated by certified veterinary pathologists. In order to stimulate farmers to submit representative fallen stock, culled animals, or euthanised diseased animals for post-mortem examination, approximately 75 percent of the costs of a post-mortem examination is financed by stakeholders of the monitoring and surveillance system. The remaining costs are covered by the submitting farmer. Based on post-mortem examinations, farmers and veterinary practitioners can approach the helpdesk for additional support and advice. GD facilitates a cadaver collection service throughout The Netherlands, which enables post-mortem examination within 24 h from Monday to Friday. These stimuli aim at ensuring that a minimal annual number of 920 randomly selected post-mortem examinations of small ruminants are conducted in order to be able to detect a new disorder with a prevalence of 1%, a test sensitivity of 50%, and a confidence-level of 99% (Table 2).

Post-mortem examinations are performed according to a standard operating procedure. Pathologists select a limited number of samples for further microscopic examination, microbiologic or toxicologic testing based on anamnesis, and macroscopic observations. In specific cases, e.g., abortion, a specific standardised sampling protocol is used [16]. All information, objective observations, and diagnostic data of the aforementioned actions are reviewed and interpreted by a veterinary pathologist to establish a probable diagnosis. Preliminary and final results are sent to the owner and associated veterinary practitioner. Results are also stored in an accessible database for the farmer and their veterinary practitioner. The database can be accessed through a specific login. In the database, an overview of results is provided, not only of post-mortem examinations but also of all laboratory tests performed at GD. Data obtained from these post-mortem examinations are also compiled and used for monitoring and surveillance purposes.

If pathologists signal diseases that are emerging, re-emerging, notifiable, or unknown, these signs are discussed with helpdesk specialists, and, if required, the national authorities will be informed. This allows a swift response and, depending on the situation, the GD veterinarian can decide to initiate active data collection.

#### 4.2.3. National and International Networks

##### Knowledge Network of Veterinary Practitioners

Since 2011, a knowledge network exists with sixteen participating veterinary practitioners from all regions of The Netherlands with an above-average interest and workload in small ruminants. The aim of this network is to exchange information regarding small ruminant health between participants and GD. GD supports one selected sheep or dairy goat farmer per participant for a period of three years. One of the small ruminant specialists of GD and their own veterinary practitioners visit these selected farms at least once a year. These farmers and veterinary practitioners share all relevant data, and in turn, have the ability to submit sheep or goats for post-mortem examination for free. In addition, participating veterinarians also are allowed to submit small ruminants for post-mortem examination at a reduced rate if they think it might be relevant for monitoring or surveillance. GD also uses this network to alert veterinarians to signals from the monitoring and surveillance system, and last but not least, this knowledge network contributes to changes and developments in this system.

##### Signalling Forum Zoonoses (SOZ)

As part of a national integrated human-veterinary risk analysis structure, SOZ was implemented in 2011. The aim of this forum is to signal and assess (potentially) zoonotic infections in humans and animals, and to serve as a platform for organisations related to human and veterinary health. The platform involves multiple organisations including GD, Wageningen Bioveterinary Research, NVWA, Municipal Health Services, Utrecht University, and the National Institute for Public Health and the Environment (RIVM), to discuss and assess collected signals on a monthly basis [86].

##### European Veterinary Surveillance Network (EVSN)

Following recommendations from an external audit on behalf of the NVWA in 2005, a European Veterinary Surveillance Network (EVSN) was established in 2008 at GD’s initiative to increase the international scope. Within this platform, Belgium, England, France, Scotland, Northern Ireland, Germany, Switzerland, and The Netherlands share knowledge about animal disease monitoring and surveillance regarding production animals. The EVSN meets annually.

#### 4.2.4. Data Analysis

An annual analysis of census data as part of the monitoring and surveillance system has been performed since 2006 (Figure 3) [71,72]. Briefly, routine census data from multiple sources are collected and sent to an external encryption company, IntoFocus Data Transformation Services (IDTS, Deventer, The Netherlands). This company encrypts all variables in the data that might link the data back to the original source, such as the farm of origin and animal identification number. To enable that data from different organisations to be linked, a corresponding encryption code is used for all datasets. Encrypted data are sent to GD and validated, aggregated, and analysed. The available data are provided by:the animal identification and registration (I&R) database operated by The Netherlands Enterprise Agency (Rijksdienst voor Ondernemend Nederland; RVO Assen, The Netherlands),the Trade Control and Expert System (TRACES) of NVWA,the database of the Dutch rendering plant Rendac (Son, The Netherlands) in which collected fallen stock is recorded,the database of GD in which post-mortem submissions to GD are recorded, andthe customer relationship management system of GD in which herd characteristics like location (2-digit postal code) and production purpose are recorded.


The data analysis provides key monitoring indicators such as animal and farm density, mortality rate, animal contact rates, and numbers and origin of imported small ruminants. Trends are analysed over a five-year period and associations between herd or flock characteristics and herd or flock health are evaluated.

### 4.3. Aggregation and Assessment

Noteworthy signs from the helpdesk and post-mortem examinations are shared and discussed at least weekly between small ruminant specialists. Information derived from all four monitoring and surveillance components is combined on a monthly basis and subsequently discussed and assessed during an interdisciplinary session with small ruminant specialists, pathologists, a toxicologist, and microbiologists with a field of expertise in bacteriology, immune diagnostics, parasitology.

### 4.4. Communication and Dissemination

In the event that a signal from one of the monitoring and surveillance components indicates a potentially urgent threat to animals or humans, GD reports immediately to the stakeholders. In case of a possible acute risk to public health, GD also informs the chairman and secretariat of the Signalling Forum Zoonoses. If necessary, neighbouring countries or EVSN members will be informed.

Quarterly consultations are held with the stakeholders during which the findings and the current small ruminant health status in The Netherlands are reported and discussed. An annual report is published with a comprehensive overview of the findings of the monitoring and surveillance system, which is publicly available.

After these stakeholder meetings, reports and information regarding relevant findings are communicated to farmers and veterinarians through various media.

### 4.5. Representativeness and Efficacy

The design of the monitoring and surveillance system is aimed at meeting its objectives: regardless of where a farmer is located, the rates for farm visits by a small ruminant specialist and for the cadaver collection service are the same, and participants of the Knowledge Network of Veterinary Practitioners are based throughout The Netherlands. The effectiveness of the system is reviewed annually, based on the numbers and geographical distribution of helpdesk consults and post-mortem examinations. In addition, the frequency with which veterinary practices contact the helpdesk and to what extent this corresponds to the expectation based on the number of unique herds or flocks per veterinary practice is examined, and independent audits take place (Figure 1).

## 5. Conclusions

This paper describes the monitoring and surveillance system of small ruminant health in The Netherlands and gives a general overview and some elaborated examples of findings since this additional voluntary system started in 2003. This system has proven highly valuable for detecting various (re)emerging and new disorders. In a continuously changing world, constant attention is needed for possible adjustments and improvements to this system, and annual checks are therefore carried out on representativeness and effectiveness. This paper provides valuable insights for those who aim to develop a comparable system.

## Figures and Tables

**Figure 1 pathogens-11-00635-f001:**
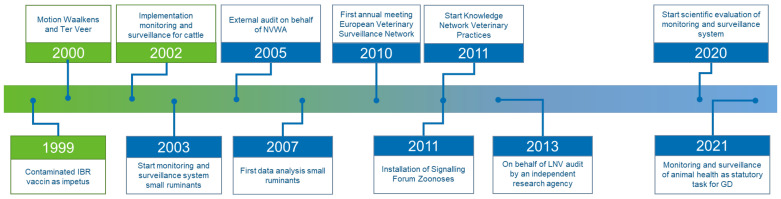
Timeline of the origin and development of the small ruminant monitoring and surveillance system in The Netherlands.

**Figure 2 pathogens-11-00635-f002:**
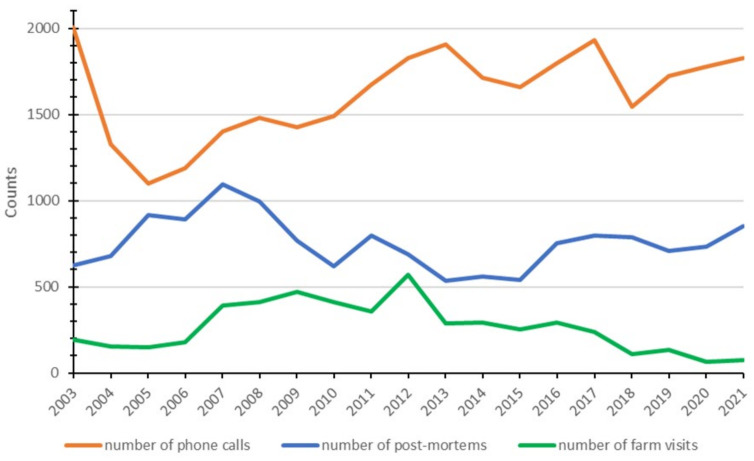
Numbers of annual phone and digital helpdesk consultations, post-mortem examinations, and farm visits received by the small ruminant monitoring and surveillance system in The Netherlands (2003–2021).

**Figure 3 pathogens-11-00635-f003:**
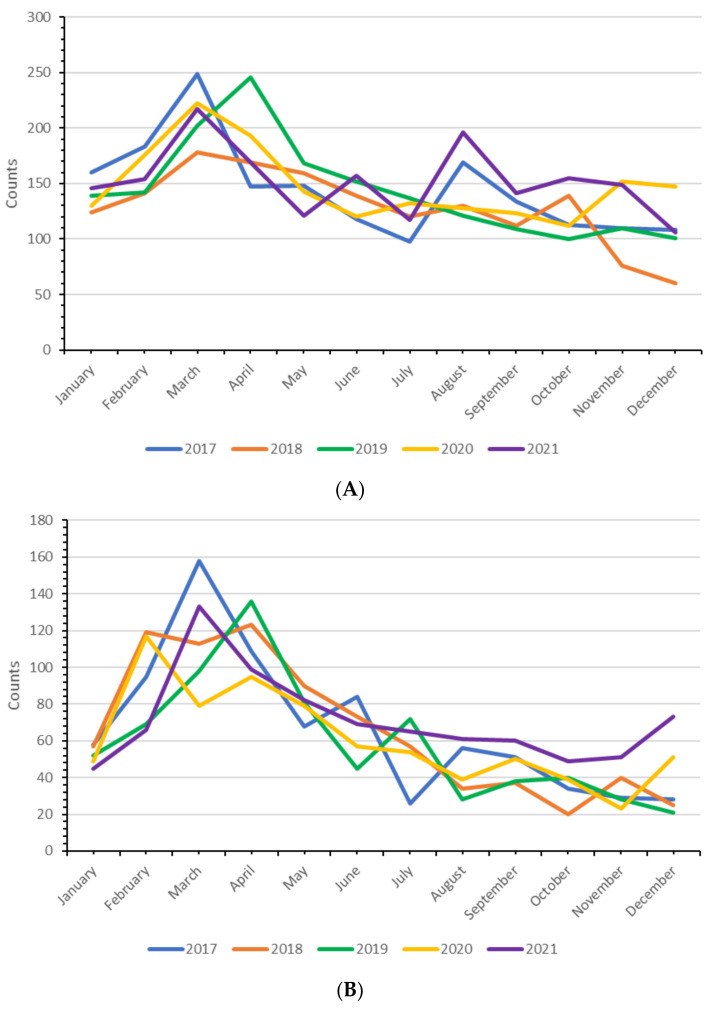
Seasonal fluctuations of helpdesk consultations (**A**) and post-mortem examinations (**B**) within the small ruminant monitoring and surveillance system in The Netherlands (2017 and 2021).

**Figure 4 pathogens-11-00635-f004:**
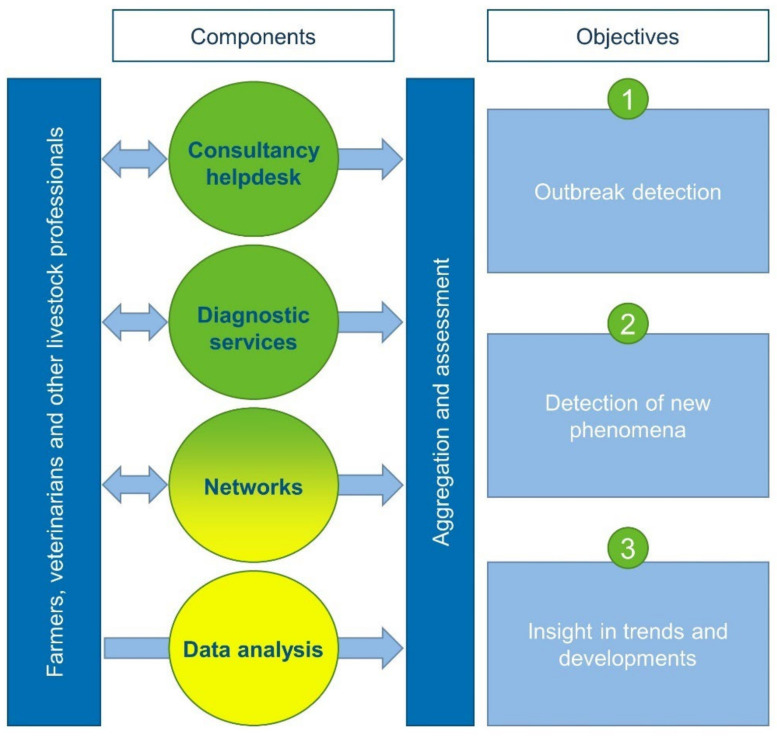
Design of the small ruminant monitoring and surveillance system in The Netherlands, showing the interaction between the field and passive (green) or active (yellow) surveillance components aiming at meeting three objectives.

**Table 1 pathogens-11-00635-t001:** Overview of selected key findings from the small ruminant monitoring and surveillance system in The Netherlands between 2003–2021. The grey blocks indicate the year or period that a specific disease was either detected or received increased attention. Monitoring and surveillance components: A. helpdesk; B. post-mortem examinations; C. laboratory investigations, in order of importance in detecting the disease in question. Copper poisoning is mainly found in sheep but in the years indicated with * 1 also confirmed in dairy goats. Anthelmintic resistance was demonstrated in: * 2 *Teladorsagia circumcincta* to ivermectin in goats, * 3 *Haemonchus contortus* to doramectin in sheep, * 4 *Haemonchus contortus* to moxidectin in sheep, and * 5 *Haemonchus contortus* to monepantel in sheep. * 6 first cases of bluetongue on a dairy goat farm. Caseous lymphadenitis confirmed * 7 in imported Solognotes, * 8 in imported Suffolks, * 9 on two dairy goat farms, * 10 in imported Lacaunes, * 11 in imported Lacaunes and on two dairy goat farms, * 12 on a sheep farm and a dairy goat farm, * 13 on a goat farm, and * 14 on a goat farm and in Drentse Heideschapen and Schoonebeekers (rare Dutch sheep breeds).

Disease/Year	2003	2004	2005	2006	2007	2008	2009	2010	2011	2012	2013	2014	2015	2016	2017	2018	2019	2020	2021	Monitoring Component (s)	References
fasciolosis																				A, B, C	[8,9,10,14]
abortion																				C, B	[8,15,16,17]
copper poisoning					* 1											* 1				A, B, C	[8]
anthelmintic resistance			* 2	* 3						* 4		* 5								A, C, B	[8,9,11,12,13,18]
osteogenesis imperfecta lambs																				A, B	[8]
coxiellosis																				B, C, A	[8,19,20,21,22,23,24,25,26,27,28,29,30,31]
severe (clostridial) metritis in goats																				B, A	[8]
bluetongue					* 6															A, C, B	[8,32,33,34,35,36,37]
caseous lymphadenitis				* 7	* 8			* 8					* 9		* 10	* 11	* 12	* 13	* 14	A, B, C	[8,38]
*Dictyocaulus filaria* in lambs																				B, C	[8]
Schmallenberg virus disease																				A, B	[8,39,40,41,42,43,44,45]
hyperoxaluria in Zwartbles sheep																				B, A	[8,46]
cerebrocortical necrosis in kids																				B, A	[8]
paresis/paralysis after Footvax^®^ vaccination																				A, B	[8]
salmonellosis dairy goat kids																				B, A, C	[8]
malignant catarrhal fever in goat																				B, C, A	[8]
enzootic nasal tumor virus in sheep																				A, B, C	[8]
osteomyelitis in kids																				A, B	[8,47]
dysbacteriosis associated diarrhoea in dairy goats																				A, B	[8]
pithomycotoxicosis in sheep																				A, B	[8,48]
jaagsiekte or ovine pulmonary adenocarcinoma																				A, B	[8]
severe outbreak of floppy kid syndrome																				A, B	[8]

**Table 2 pathogens-11-00635-t002:** Statistical model designed by GD to calculate the required number of randomly selected post-mortem examinations to detect disorders with a reliability of 99% at different prevalences (in the studied population) and different sensitivities.

	Sensitivity (%)			
Prevalence (%)	25	50	75	90
0.1	18,000	9000	6000	5000
0.5	3280	1640	1230	1020
1.0	1840	920	613	511
5.0	360	180	120	100
10.0	176	88	59	49

## Data Availability

An overview of results as presented in this paper is published in the small ruminant half-yearly or annual monitoring and surveillance reports (Rapportage Monitoring Diergezondheid Kleine Herkauwers) and the cited references.

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
