# Peer review of "Monitoring and Surveillance of Small Ruminant Health in The Netherlands"

_pathogens, 2022, doi:10.3390/pathogens11060635_

Round 1
Reviewer 1 Report
Thank You for the corrections made to the article.
I would like to highlight a few minor points that need to be improve:
- the abbreviation VLA is not explained in the text (line 141)
- "Typhimurium" should be written in italics (line 154)
- chapter 2.2 "schmallenberg" as proper name should be capitalized
- "van den" should be standardized in the bibliography. It is written once in capitals and once in small letters, even for the same person, e.g. items 22, 48 and 85- from a small letter, positions 24, 25 and 40 from a capital letter
- items 68 and 85 need to be improve. They have a duplicate year, journal issue and page number.
Author Response
Please, see attachment.

Reviewer 2 Report
The article “Monitoring and surveillance of small ruminant health in the Netherlands” by Dijkstra et al., sent for consideration at MDPI Pathogens, is an original article that has the intention to describe and discuss the current monitoring and surveillance system ongoing in the Netherlands.
The authors should consider the paper as a “Review” instead of “original research” and change the current structure to improve readability and comprehension. Additionally, the text would benefit from English editing in clarity and grammar.
Issues to be addressed:
After the title “overview of findings” the line numbering is missing, hindering the review process.
There are several spelling errors throughout the manuscript: dection (line 27); Fluctiations (line 116); disease specific (line 57 – should have a hyphen); schmallemberg virus (section “Elaborated examples of three findings’ – must be with must start with a capital letter); highy (section “detection of an unknown disease: the case of Schmallemberg virus); and many other mistakes hard to be referenced due to the lack of line numbers (spead instead of spread; or coöperation).
Also, it is necessary to standardize the writing. For example, you have “post mortem” and “post-mortem” across the manuscript.
It seems that the submitted text is the first draft of the paper without any corrections or proofreading.
INTRODUCTION:
Most of the information presented in the section “Materials and methods” should be inserted in the introduction section to clarify to the reader how the monitoring and surveillance system in the Netherlands works.
In the way the paper is structured, it is hard to understand the paper and, especially, the result section.
RESULTS:
Line 92: Figure 1 – Please, insert the legend for some abbreviations that didn’t appear in the text until now.
Lines 121 – 181: This paragraph is too long. I suggest splitting between findings/illnesses. I also recommend that this paragraph be placed after figure 4.
After line 181: Figure 4 should be designated as a table. Also, Standardize the caption. Numbers appear before and after the description.
Sub-section “ 2.2.1 Detection of the incursion of a known disease: the case of bluetongue”: “The bluetongue associated mortality in sheep was estimated to be 27,000 in the second half of 2007” – Please, if possible, insert the total number of animals affected.
Sub-section “2.2.3 Monitoring trends and developments: the case of Coxiella burnetii infections” – Please, in the headline, underline Coxiella burnetii.
MATERIALS AND METHODS:
Most parts of this section should be transferred to the introduction section to improve clarity on how the monitoring and surveillance system works in the Netherlands with a consequent reflex on the comprehension of the result section.
Only how the data were curated to obtain the figures and tables must be described here.
Author Response
Please, see attachment.

Reviewer 3 Report
The authors of the manuscript entitled “Monitoring and surveillance of small ruminant health in the 2 Netherlands”, describe a voluntary system in the Netherlands in small ruminants that was developed for the monitoring and surveillance of small ruminant health. In addition, they offer an overview of key findings (+ examples) over a period of almost 20 years (since 2003). In my opinion, the manuscript is very well written, with a lot of useful information for other countries which are willing to implement a similar system. The Netherlands is well known for its voluntary programs, especially those used to eradicate infectious diseases. There are some small changes to be made and a few things to be added.
In my opinion, it’s important to mention the current number of small ruminants (sheep/goats) present in the Netherlands. Maybe a graph with the fluctuation of the total number of small ruminants in the period 2003-2022 can be added. This may be related to some of the key findings.
Line 27 : Detection, not dection
Line 65-66 : I think is better to use this sentence “to the use of IBR vaccine that was contaminated with BVD-virus type 2”
Line 71: without of
Figure 1: All the abbreviations should be explained. Not only in text.
Line 127: The citations 14-18 are not good; In the text, the authors are talking about parasites, but the citations are related to Coxiella
I suggest to the authors present these results at conferences related to small ruminants (ex. European College of Small Ruminant Health Management Conference).
Author Response
Please, see attachment.

This manuscript is a resubmission of an earlier submission. The following is a list of the peer review reports and author responses from that submission.
Round 1
Reviewer 1 Report
First, thank you for your interesting work related to monitoring the health of small ruminants.
In general, in my opinion, the structure of the article is correct, except mainly for the discussion. However, I have a problem with its scientific overtones. Can an article describing a monitoring system that has been in place since 2003 be considered a research article? Isn't it some kind of review article or another type?
However, setting that aside, I have a few comments about the article.
It would be good to mention the four surveillance components in abstract.
Figure 1.- If the text mentions that the whole system starts from 2003, wouldn't it be good to mark it with a different color on the diagram, or start the diagram with it?
There are a few typos, such as "diary goats" instead of "dairy goats" in the discussion, or using the phrase "small ruminant" instead of "small ruminants".
I have a problem reading this discussion in general, as it is initially an expansion of what is included in the introduction, then a description of the results obtained and possible causes (discussion of figures), and finally a description of the diseases. Not a typical discussion structure, I think it is a chapter worth refining.
Thanks again for the interesting perspective on the health monitoring system in small ruminants.
Reviewer 2 Report
I appreciate that the article links science with practice. The aims - " 1) early detection of outbreaks of designated animal diseases, 2) early detection of yet unknown disease conditions, and 3) insight in trends and developments" - are of really high importance.
Reviewer 3 Report
Thank you for submitting your article. I am confused on the purpose of the paper. I understand it was to describe the monitoring system and give an overview of examples. However, where is the impact of that information? Can the information from the voluntary system be used in a different way to evaluate how well the system is doing? I am not sure how this information can help producers or health professionals.
Abstract:
The second to last line does not need a comma after the word and.
Introduction:
Disease’s should be disease
The first sentence is awkward, please reword.
The first paragraph, last sentence is very long. Please break up into two sentences.
Results:
2.2.1: fourth line down. The word “was” should be “were”.
Some of the information in your results section should go in discussion as it is more discussion based
Materials and methods:
Please place these before the results section
Tables and figures:
The blue writing on blue background for figure 1 is hard to read, please change the coloring.